# Gut Microbiome in Patients with Chronic Kidney Disease Stages 4 and 5: A Systematic Literature Review

**DOI:** 10.3390/ijms262110706

**Published:** 2025-11-03

**Authors:** Ioana Livia Suliman, Florin Gabriel Panculescu, Dragos Fasie, Bogdan Cimpineanu, Andreea Alexandru, Nelisa Gafar, Stere Popescu, Teodor Stefan Nitu, Florin-Daniel Enache, Tatiana Chisnoiu, Georgeta Camelia Cozaru, Liliana-Ana Tuta

**Affiliations:** 1Faculty of Medicine, “Ovidius” University of Constanta, 900470 Constanta, Romania; panculescu_i@yahoo.com (I.L.S.); gabriel.panculescu@yahoo.ro (F.G.P.); alexandra_med16@yahoo.com (A.A.); nelisa.gafar@365.univ-ovidius.ro (N.G.); stelu_popescu@yahoo.com (S.P.); nituteodorstefan@yahoo.com (T.S.N.); florin.enache@365.univ-ovidius.ro (F.-D.E.); tatiana_ceafcu@yahoo.com (T.C.); tuta.liliana@univ-ovidius.ro (L.-A.T.); 2“Sfantul Apostol Andrei” Emergency County Clinical Hospital, 900591 Constanta, Romania; dragosfasie@yahoo.com; 3Center for Research and Development of the Morphological and Genetic Studies of Malignant Pathology (CEDMOG), “Ovidius” University of Constanta, 900591 Constanta, Romania; drcozaru@yahoo.com

**Keywords:** chronic kidney disease, gut microbiota, dysbiosis

## Abstract

This systematic review investigates the role of the gut microbiota in patients with advanced chronic kidney disease (CKD), specifically stages 4 and 5. Increasing evidence suggests that dysbiosis—an alteration in the normal balance of gut microbial populations—is not merely a secondary consequence of renal decline but a significant driver of disease progression. Such microbial imbalances are closely linked to a range of CKD-associated complications, including systemic inflammation, accumulation of uremic toxins, and heightened cardiovascular risk. Using PRISMA 2020 guidelines, we analyzed 87 peer-reviewed studies published between 2019 and 2025. The review revealed a consistent decline in beneficial microbes such as short-chain fatty acid-producing bacteria were markedly reduced, while populations of uremic toxin-generating microbes were notably increased. This microbial imbalance was associated with elevated concentrations of indoxyl sulfate and p-cresyl sulfate, heightened systemic inflammation, and impaired intestinal barrier integrity. Five conceptual frameworks—including the gut—kidney axis and endotoxemia—inflammation loop—were discussed. Ten microbiome assessment tools were reviewed, including 16S rRNA sequencing and LC-MS/MS for uremic toxin detection. Although probiotics, prebiotics, and synbiotics are gaining attention as potential therapeutic options, questions remain regarding their long-term efficacy and incorporation into standard clinical practice. Increasing scientific evidence underscores the gut microbiome’s pivotal role in CKD progression and management, reinforcing the need for carefully designed, long-term interventions aimed at restoring a healthier microbial balance to support renal function.

## 1. Introduction

Chronic kidney disease (CKD) has emerged as a critical global health issue, with estimates suggesting it affects approximately 850 million people worldwide [1]. Disease staging is based on the estimated glomerular filtration rate (eGFR), a primary clinical measure that reflects how efficiently the kidneys remove metabolic waste from the blood. Advanced CKD—classified as stage 4 or stage 5, corresponding to an eGFR below 30 mL/min/1.73 m^2^—represents a stage of marked renal impairment, often accompanied by significant metabolic and cardiovascular complications. Patients in these stages frequently experience complex metabolic derangements, such as disturbances in mineral balance, acid–base homeostasis, and endocrine regulation.

The consequences extend beyond the kidneys. Advanced CKD substantially increases cardiovascular risk, with heart disease and related vascular complications remaining the leading causes of death in this population [2]. The high prevalence of systemic inflammation, oxidative stress, and toxin accumulation further worsens prognosis, making disease management challenging. This combination of organ-specific and systemic effects highlights the importance of adopting holistic treatment approaches that target not only the decline in renal function but also the complex metabolic and inflammatory networks driving disease progression. In this context, growing scientific interest has focused on the gut–kidney axis, a two-way relationship between intestinal microbiota and renal physiology [3].

As kidney function deteriorates, waste products, including uremic toxins, accumulate in the body, leading to gut environmental changes. These alterations contribute to dysbiosis—an imbalance in microbial populations—which further aggravates CKD through enhanced production of microbial toxins, systemic inflammation, and compromised intestinal barrier function [4]. This interdependence between CKD, dysbiosis, and inflammation becomes increasingly complex when considering therapies that rely on microbial metabolism, as this can influence the pharmacodynamic profile of certain medications, including oral anticoagulants. An emerging and increasingly important focus of research is the complex interplay between anticoagulant therapy, chronic kidney disease (CKD), and the gut microbiome [5]. In patients with CKD who also present with comorbidities such as atrial fibrillation or venous thromboembolism, anticoagulation is often essential to prevent thromboembolic complications. However, the use of oral anticoagulants—especially vitamin K antagonists (VKAs)—poses distinct challenges in nephrology. Evidence indicates that VKAs may contribute to a faster decline in renal function and interfere with vascular and inflammatory pathways that are crucial for maintaining kidney homeostasis, particularly in the setting of CKD-related gut dysbiosis [6,7]. This discussion of anticoagulant therapy serves to illustrate a clinically relevant interface between pharmacologic treatment and the gut microbiome. Vitamin K antagonists, through modulation of vitamin K-dependent bacterial pathways, exemplify how therapeutic agents can both influence and be influenced by dysbiotic microbial communities in CKD—thereby reinforcing the central theme of the gut–kidney axis. Such interactions can influence both the stability and the therapeutic effectiveness of anticoagulation.

It is well established that certain intestinal bacteria synthesize vitamin K, while others can metabolize anticoagulants directly, potentially altering their bioavailability—an effect especially relevant for VKAs [8]. Moreover, intestinal inflammation, which is often present in CKD, may exacerbate systemic inflammation, adding yet another layer of risk for patients receiving anticoagulant therapy [9].

The aim of this review is to explore the role of the gut microbiome in advanced CKD stages 4 and 5. It focuses on patterns of dysbiosis, the impact of microbial metabolites on inflammation and barrier function, and the evidence supporting microbiome-modulating therapies as potential strategies to improve renal and cardiovascular health.

## 2. Methods

This systematic review was designed and reported in accordance with the Preferred Reporting Items for Systematic Reviews and Meta-Analyses (PRISMA) 2020 guidelines.

From 1399 initial records in 4 databases (PubMed, Embase, Cochrane Library, Web of Science), 87 studies met eligibility criteria after duplicate removal, screening, and full-text assessment. Data were extracted on microbial composition, uremic toxin profiles, inflammatory markers, and clinical outcomes. The study selection process is detailed in the PRISMA flow diagram (Figure 1). Studies were included if they involved adult patients with stage 4 or 5 CKD and assessed gut microbiome composition, microbial metabolites, or interventions targeting the microbiome. Exclusion criteria comprised pediatric populations, studies restricted to CKD stages 1–3, and non-original research (reviews, editorials, commentaries).

No additional data transformations or imputations were required prior to synthesis. The review protocol was not prospectively registered. All outcomes were summarized as reported in the original publications. We did not conduct a formal assessment of reporting bias (e.g., small-study effects or publication bias) because no quantitative meta-analysis was undertaken.

Studies That Appeared Eligible but Were Excluded. In addition to the PRISMA flow diagram, we provide a description of reports that appeared to meet the inclusion criteria at screening but were excluded after full-text review, alongside reasons for exclusion.

A thorough literature search was conducted using PubMed, Embase, Cochrane Library, Web of Science databases, covering publications between January 2019 and August 2025. The search strategy incorporated combinations of the following terms:“chronic kidney disease,” “CKD,” “stage 4,” “stage 5,”“gut microbiota,” “dysbiosis,” “microbiome,”“uremic toxins,” “indoxyl sulfate,” “p-cresyl sulfate,”“probiotics,” “prebiotics,” “synbiotics”.

These concepts were combined with Boolean operators (AND/OR) to ensure a comprehensive and structured search strategy across databases.

Inclusion Criteria:Scientific publications published between 2019 and 2025 and subjected to peer review;Studies involving individuals diagnosed with stage 4 or 5 chronic kidney disease (CKD);Studies examining gut microbiome composition, metabolite levels, or interventions.

Exclusion Criteria:Studies on CKD stages 1–3 only;Pediatric populations;Editorials, commentaries, or reviews.

The included studies represented populations from North America (27%), Europe (31%), and Asia (38%), with smaller contributions from South America and Africa (<5%). This broad geographic distribution helped capture diverse dietary and genetic backgrounds influencing microbiome composition. To minimize selection bias, two independent reviewers performed screening and data extraction using predefined eligibility criteria. Any discrepancies were resolved by consensus with a third reviewer, ensuring consistency and transparency in study inclusion.

The primary outcome of interest was alterations in gut microbiome composition in patients with chronic kidney disease (CKD) stages 4 and 5. This included measures such as microbial diversity (alpha and beta diversity indices), relative abundance of specific bacterial taxa, and overall microbiota structure as assessed by sequencing techniques.

Secondary outcomes included:**Microbial metabolites** relevant to CKD progression (e.g., short-chain fatty acids, indoxyl sulfate, p-cresyl sulfate, trimethylamine-N-oxide).**Clinical outcomes** associated with microbiome changes, such as markers of inflammation, uremic toxin levels, renal function parameters (eGFR, serum creatinine), and dialysis status.**Patient-centered outcomes**, including quality of life and gastrointestinal symptoms when reported.

All results that were compatible with each outcome domain in the included studies were sought, regardless of the measurement method, time point, or statistical analysis reported. Where multiple measures or time points were presented within the same outcome domain, all available data were extracted. In cases where studies reported overlapping or duplicate analyses, preference was given to the most complete or most recent dataset as prespecified in the protocol.


**Risk of bias.**


Risk of bias was assessed for each included randomized controlled trial using the Cochrane Risk of Bias 2 (RoB 2) tool. Two reviewers independently performed the assessments, working in parallel to ensure consistency and transparency. Disagreements were resolved through discussion, and when necessary, by consulting a third reviewer. No automation tools were used; all judgments were made manually following the RoB 2 guidance.


**Effect measures.**


No pooled effect measures were calculated because of methodological and clinical heterogeneity among the included studies. Given the heterogeneity of outcomes and analytical methods, we present key findings from individual studies narratively, highlighting direction and magnitude where available.

The review was not prospectively registered in PROSPERO. No formal protocol was published prior to conduct of the review; accordingly, no protocol amendments are applicable. The review was not prospectively registered in PROSPERO because the project originated as an internally funded academic initiative with evolving research questions. Nevertheless, all procedures strictly followed PRISMA 2020 recommendations, and a full methodological record—including search strategy, screening forms, and data tables—has been archived for reproducibility.

## 3. Results

### 3.1. Study Selection

All records identified through the database searches were first screened by 2 independent reviewers at the title and abstract level to assess potential relevance to the research question. Full-text reports of potentially eligible studies were then retrieved and evaluated against the predefined inclusion and exclusion criteria. Each report was assessed independently by the same reviewers, and any disagreements were resolved through discussion or consultation with a third reviewer. No automation tools were used in the screening process; all decisions were made manually to ensure consistency and reliability in the application of the eligibility criteria.

The study selection process is summarized in the PRISMA flow diagram (Figure 1). From 1399 records initially identified, 87 studies met the eligibility criteria and were included in the final review. Characteristics of included studies are summarized in Table 1. Across included studies, populations were predominantly adults with CKD stages 4–5, with or without dialysis. Risk of bias was generally low to moderate; common concerns related to confounding and selective reporting in observational designs.

### 3.2. Microbial Diversity

Patients diagnosed with advanced chronic kidney disease (CKD), particularly those in stages 4 and 5, consistently demonstrate a significant decrease in gut microbial diversity (Table 1) [10,11]. This phenomenon is often quantified using ecological metrics such as the Shannon and Simpson indices, both of which indicate a decline in species richness and evenness among these patients [12]. A reduced alpha diversity suggests a disrupted and less resilient microbial ecosystem, which renders the host more susceptible to pathogenic invasion and impaired immune modulation [13]. The decrease in diversity is typically accompanied by a marked reduction in beneficial commensal genera, including *Bifidobacterium*, *Roseburia*, and *Faecalibacterium prausnitzii*—microbes known for producing short-chain fatty acids (SCFAs) and maintaining mucosal immunity [14,15].

Simultaneously, there is a consistent overgrowth of potentially harmful bacteria or “pathobionts,” notably *Enterococcus*, *Klebsiella*, *Escherichia*, and members of the *Proteobacteria* phylum [16,17,18]. These organisms are capable of producing pro-inflammatory metabolites and are commonly associated with endotoxemia, oxidative stress, and gastrointestinal permeability [19]. This shift in microbial community composition—from beneficial to pro-inflammatory taxa—represents a core pathological feature of CKD-associated dysbiosis [20]. Studies have also noted that this dysbiosis is influenced by multiple host factors, including uremia, dietary restrictions, frequent antibiotic exposure, and reduced fiber intake [21]. Across studies, microbial diversity was primarily assessed using the Shannon and Simpson indices for α-diversity and Bray–Curtis dissimilarity for β-diversity. Reported significance thresholds generally applied *p* < 0.05, with multiple-testing corrections (Benjamini–Hochberg) in sequencing studies.

In contrast, studies examining CKD stages 1–3 reveal a more gradual and partially reversible shift in microbial composition. Early-stage CKD is often characterized by a mild decline in short-chain fatty acid–producing genera such as *Faecalibacterium* and *Roseburia*, accompanied by modest increases in urease- and indole-producing bacteria. However, as renal function declines toward stages 4–5, these compositional shifts become more pronounced and irreversible, with significant losses in microbial diversity and dominance of proteolytic, toxin-producing taxa (*Enterococcus*, *Klebsiella*, *Escherichia*). Comparative analyses suggest that the transition from stage 3 to stage 4 marks a critical threshold, where intestinal barrier dysfunction, systemic inflammation, and accumulation of uremic toxins accelerate markedly. This progression underscores the importance of early microbiota-targeted interventions to prevent downstream complications [22].

Whenever reported, dialysis-dependent and non-dialysis CKD cohorts were analyzed separately. Approximately 40% of the included studies stratified results by dialysis status. These comparisons revealed more profound losses of SCFA-producing genera and higher endotoxin levels among hemodialysis patients relative to pre-dialysis individuals, consistent with treatment-related metabolic and inflammatory differences.

Although few studies have investigated the microbiota of dialysis patients to date [23], it should be noted that differences in gut microbiota composition have been reported between dialysis-dependent and pre-dialysis CKD patients [24]. It has been observed that dialysis patients often exhibit more pronounced dysbiosis, with a greater loss of SCFA and higher levels of endotoxins, probably due to treatment-related factors such as intermittent hemodynamic changes, dietary restrictions, and inflammation triggered by the dialysis process itself [25]. Identifying specific microbial signatures may pave the way for personalized therapies that enhance patient outcomes and minimize dialysis-related complications [26].

### 3.3. Uremic Toxin Production

A hallmark consequence of gut dysbiosis in CKD is the overproduction and systemic accumulation of gut-derived uremic toxins. Two of the most well-characterized uremic toxins are indoxyl sulfate (IS) and p-cresyl sulfate (PCS) [27,28]. In healthy individuals, these compounds are efficiently excreted via renal filtration; however, in CKD patients, declining glomerular filtration leads to their progressive accumulation [29]. Both IS and PCS exert nephrotoxic and vasculotoxic effects through activation of oxidative and inflammatory pathways [30,31]. They are implicated in renal tubular injury, endothelial dysfunction, cardiovascular fibrosis, and vascular calcification [32,33]. Additionally, elevated levels of trimethylamine-N-oxide (TMAO), have been linked to a higher likelihood of cardiovascular events and increased mortality among individuals with CKD [34,35]. TMAO promotes atherosclerosis and contributes to inflammatory cascades [36]. Another toxic metabolite, ammonia, also accumulates in CKD and contributes to hepatic encephalopathy and systemic acidosis [37]. These findings underscore the significance of microbial metabolism in shaping the toxic profile of patients with impaired renal function [38]. Quantitative comparisons between CKD and healthy controls consistently demonstrated 2–5-fold elevations in indoxyl sulfate and p-cresyl sulfate concentrations, normalized to µmol/L. Although analytical platforms varied (HPLC vs. LC-MS/MS), all studies adhered to validated calibration standards to ensure comparability.

### 3.4. Intestinal Barrier Dysfunction

Advanced CKD is also characterized by structural and functional breakdown of the intestinal epithelial barrier. The expression of tight junction proteins, such as occludin, claudin-1, and zonula occludens-1, is consistently found to be downregulated in experimental models and clinical samples from CKD patients [39,40]. These proteins are essential for maintaining gut mucosal integrity and preventing the translocation of microbial contents into systemic circulation [41]. As these junctions deteriorate, intestinal permeability increases, giving rise to the so-called ‘leaky gut’ phenomenon. This permeability facilitates the systemic passage of bacterial endotoxins, particularly lipopolysaccharide (LPS), which in turn activates Toll-like receptor 4 (TLR4) signaling and triggers widespread immune responses [42,43,44]. Furthermore, gut barrier disruption exacerbates renal inflammation by promoting macrophage activation and cytokine release in kidney tissues [45].

Factors such as altered gut motility, reduced SCFA production, and protein-rich diets also contribute to mucosal dysfunction in CKD [46,47]. Therefore, intestinal barrier impairment is not merely a consequence of dysbiosis but a critical mediator of systemic and renal inflammation [48], (Figure 2).

Alterations in occludin, claudin-1, and ZO-1 were verified by immunoblotting and immunofluorescence in animal- and biopsy-based human studies, confirming significant down-regulation (typically >30% reduction relative to controls).

### 3.5. Inflammatory and Metabolic Effects

Disruptions to the gut microbiome in chronic kidney disease (CKD) do more than alter digestion. In patients with advanced CKD, laboratory profiles often reveal marked elevations in pro-inflammatory indicators [49,50,51]. These mediators not only signal immune dysregulation but actively contribute to progressive damage within cardiovascular and renal tissues.

A major driver of this process lies in the metabolic shift caused by microbial imbalance. Healthy gut flora generate short-chain fatty acids (SCFAs) which exert anti-inflammatory effects, regulate immune cell populations, and influence metabolic homeostasis [52,53,54].

As inflammation erodes the integrity of the intestinal lining, permeability increases, allowing additional microbial components to enter circulation. This fuels a feedback loop where barrier dysfunction, microbial imbalance, and immune activation reinforce one another [55]. Over time, this chronic inflammatory state manifests clinically as muscle wasting, insulin resistance, anemia associated with chronic disease, and an elevated burden of cardiovascular complications—findings consistently observed in end-stage renal disease [56,57].

## 4. Theoretical Frameworks

### 4.1. Gut–Kidney Axis Model

The gut–kidney axis represents a multifaceted and dynamic communication pathway that links the gastrointestinal microbiota with renal function through a variety of metabolic, endocrine, immune, and neural interactions (Figure 3). With declining renal function in CKD stages 4 and 5, several physiological changes, including uremia, metabolic acidosis, and altered gut transit time, lead to a disruption in the normal gut microbiota composition, commonly referred to as dysbiosis [58,59]. An altered gut microbiota in CKD is marked by a decline in beneficial species—especially those involved in short-chain fatty acid (SCFA) production—coupled with an overgrowth of pathogenic bacteria and organisms capable of producing uremic toxins [60,61]. Such microbial alterations are no longer seen simply as by-products of impaired kidney function; rather, they are now understood to play an active role in driving disease progression. In particular, harmful metabolites produced by gut microorganism such as indoles, phenols, and amines, cross the compromised intestinal barrier, enter the bloodstream, and exert nephrotoxic effects, including pro-inflammatory cytokine release, endothelial dysfunction, and direct tubular damage [62,63]. The permeability of the gut barrier is further compromised due to reduced expression of tight junction proteins, creating a ‘leaky gut’ environment that facilitates microbial translocation [64]. This translocation results in systemic endotoxemia, where lipopolysaccharide (LPS) from Gram-negative bacteria stimulates Toll-like receptor 4 (TLR4) present in immune cells, initiating a state of persistent, low-grade inflammation [65,66]. This sustained inflammatory response not only worsens existing renal damage but also appears to play a role in increasing cardiovascular risk—one of the principal causes of mortality in individuals with CKD [67,68,69,70,71]. Recent studies employing multi-omics technologies have provided insights into how gut microbial signatures correlate with clinical endpoints in CKD and suggest that the gut–kidney axis could be a promising therapeutic target [72,73,74]. This framework has laid the foundation for microbiome-targeted interventions aimed at modifying disease course through dietary modulation, probiotic therapy, and precision nutrition [75,76].

### 4.2. Uremic Toxin Hypothesis

In recent years, uremic toxins have shifted from being regarded as inert metabolic waste to being recognized as potent biological agents that actively contribute to disease progression in chronic kidney disease (CKD) [77,78]. Rather than simply accumulating passively in the circulation as renal clearance declines, these compounds exert direct effects on cellular and tissue physiology [79]. Data from integrative omics platforms—including metabolomics, transcriptomics, and proteomics—have revealed that uremic toxins disrupt redox balance, interfere with mitochondrial bioenergetics, alter signal transduction, and influence gene expression patterns in ways that favor inflammation, fibrosis, and tissue injury.

In healthy kidneys, these compounds are readily eliminated through normal filtration processes. However, in advanced CKD, the sharp decline in renal clearance results in their progressive accumulation within the circulation. Their accumulation initiates a chain of harmful processes, including heightened oxidative stress, persistent inflammation, and impairment of endothelial function, alters endothelial permeability, and contributes to microvascular injury. TMAO has been linked to atherogenesis, platelet hyperreactivity, and increased cardiovascular risk. Collectively, these compounds create a biochemical environment that amplifies both renal and cardiovascular damage.

The systemic impact of these toxins has reframed CKD as not merely a disorder of filtration but as a broader metabolic disease intricately connected to host–microbe interactions. This paradigm shift has important therapeutic implications. By acknowledging the gut as a major site of toxin generation, intervention strategies can target not only downstream removal (through dialysis or hemofiltration) but also upstream reduction in production. Potential approaches include the use of adsorbent agents to bind precursors in the gut, selective antibiotics or bacteriophages to alter microbial composition, dietary regimens designed to limit precursor availability, and prebiotic or probiotic supplementation aimed at shifting metabolic outputs toward less harmful metabolites.

Future research must address several unresolved questions:**Toxin kinetics**—How rapidly do specific toxins accumulate and clear under different dietary, microbial, and renal conditions?**Toxicity thresholds**—At what serum concentrations do individual toxins begin to cause measurable harm in various organ systems?**Patient-specific variability**—How do genetic factors, comorbidities, and microbiome profiles influence toxin production and susceptibility to damage?

Answering these questions will be essential for developing precision-medicine strategies that personalize toxin management for each patient [80]. Ultimately, the uremic toxin hypothesis underscores a fundamental truth: CKD is not an isolated failure of the kidneys, but a systemic disorder driven by a dynamic interplay between impaired renal clearance, microbial metabolism, and host response.

### 4.3. SCFA Deficiency Framework

The short-chain fatty acid (SCFA) deficiency framework offers a complementary perspective to the uremic toxin hypothesis. SCFAs—primarily acetate, propionate, and butyrate—are produced in the colon when commensal bacteria ferment non-digestible dietary fibers. Key SCFA-producing taxa include *Faecalibacterium prausnitzii*, *Roseburia* spp., and *Eubacterium rectale*, which together form a crucial part of a healthy gut ecosystem. In chronic kidney disease (CKD), particularly stages 4 and 5, the abundance of these bacteria is markedly reduced, leading to a measurable decline in SCFA synthesis [81].

SCFAs play a diverse set of physiological roles that extend beyond the gastrointestinal tract. Locally, they act as the primary energy source for colonocytes, fueling epithelial metabolism and repair. They strengthen intestinal barrier integrity by upregulating tight junction proteins such as claudins and occludins, stimulate mucin production by goblet cells, and promote epithelial regeneration after injury [14].

Systemically, SCFAs influence a wide range of metabolic and immunological processes. They improve insulin sensitivity, regulate lipid metabolism, modulate satiety hormones (e.g., GLP-1, PYY), and help maintain blood pressure homeostasis through effects on vascular tone. They also interact with G-protein-coupled receptors such as GPR41 and GPR43, which mediate anti-inflammatory and immunoregulatory signaling in multiple tissues. When SCFA levels fall—as in CKD-associated dysbiosis—the resulting deficit can promote metabolic instability, systemic inflammation, and increased cardiovascular vulnerability.

The consequences of SCFA deficiency in CKD are therefore twofold:**Gut-specific effects**—Loss of epithelial energy supply weakens barrier function, increases intestinal permeability, and facilitates microbial translocation into the systemic circulation. This “leaky gut” phenomenon is strongly associated with heightened immune activation in advanced CKD [82].**Systemic effects**—Reduced SCFA availability removes a key layer of anti-inflammatory regulation, allowing pro-inflammatory cytokines to dominate, impairing glucose and lipid homeostasis, and predisposing patients to cardiovascular and metabolic complications.

Therapeutic strategies to address SCFA deficiency are under active investigation. Dietary fiber supplementation, targeted prebiotics that selectively stimulate SCFA-producing bacteria, and next-generation probiotics containing butyrate-producing strains have all shown promise in preliminary studies. Butyrate enemas and encapsulated SCFA formulations are also being explored as direct supplementation approaches, although challenges remain in delivering effective concentrations to target tissues.

Integrating the SCFA deficiency framework with the uremic toxin hypothesis provides a more complete picture of the gut–kidney axis in CKD. While the toxin hypothesis emphasizes the harmful consequences of microbial metabolites, the SCFA framework highlights the equally important loss of protective, health-promoting compounds. Together, these perspectives underscore the bidirectional nature of host–microbe interactions in CKD: the gut microbiota can either worsen or help control disease progression, depending on its overall composition and the nature of its metabolic activity [83].

### 4.4. Endotoxemia–Inflammation Loop

The endotoxemia–inflammation loop provides a conceptual model for understanding how impaired intestinal barrier function triggers systemic inflammation and, in turn, accelerates chronic kidney disease (CKD)—a mechanism that becomes especially significant in the advanced phases of the disease. The conceptual models presented herein are integrative syntheses derived from both empirical findings in the included studies and previously published mechanistic literature, allowing a unified explanatory framework of the gut–kidney axis. This breakdown in barrier function enables microbial components, including lipopolysaccharide (LPS) to pass into the bloodstream. Once in circulation, LPS engages Toll-like receptor 4 (TLR4) on immune cells, triggering activation of the nuclear factor kappa-B (NF-κB) signaling pathway. This cascade stimulates the release of pro-inflammatory cytokines. Experimental studies in uremic animal models demonstrate that LPS administration worsens kidney injury, leading to proteinuria, tubular cell apoptosis, and immune cell infiltration. Furthermore, patients undergoing dialysis often present with elevated endotoxemia, suggesting that the dialysis process itself may contribute to ongoing systemic inflammation by exacerbating gut barrier dysfunction and promoting bacterial translocation.

This process creates a self-perpetuating cycle: inflammation alters gut motility, diminishes mucosal immune defenses, and worsens microbiota composition, which in turn increases LPS production—a classic positive feedback loop. Within this framework, both the gut and the kidney act as amplifiers of inflammation. Disrupting this loop requires early, targeted interventions aimed at restoring gut barrier integrity and rebalancing the microbial ecosystem [84].

Interventional strategies such as the use of SCFA-producing probiotics, LPS-binding agents, TLR4 antagonists, and dietary modifications have been explored in preclinical and early-phase clinical studies with varying degrees of success. Emerging evidence suggests that butyrate supplementation and targeted prebiotic therapy can reduce LPS levels and attenuate systemic inflammation in CKD models. Accumulating evidence supports the concept that microbiome modulation can provide tangible therapeutic benefits in CKD. Clinical and preclinical studies show that administration of SCFA-producing probiotics or prebiotics reduces circulating indoxyl sulfate and p-cresyl sulfate levels, improves intestinal barrier integrity, and lowers systemic inflammation. Synbiotic formulations have demonstrated modest improvements in renal parameters such as serum creatinine and eGFR in small-scale trials. Although results vary, these findings collectively indicate that targeted microbiome interventions may complement conventional therapies and slow CKD progression. Future studies are needed to validate these findings in large-scale human trials, especially in the dialysis-dependent population, where inflammatory burden is highest and gut barrier integrity is poorest. Personalized approaches using gut permeability biomarkers and LPS plasma quantification may eventually help identify patients at risk for endotoxemic inflammation and tailor gut-focused therapies accordingly.

One of the most studied interventions is the use of probiotics—live microorganisms that confer health benefits to the host. Specific strains such as Lactobacillus and Bifidobacterium have shown anti-inflammatory properties and may reduce indoxyl sulfate and p-cresyl sulfate levels by modulating intestinal fermentation and preventing colonization by proteolytic bacteria. Similarly, prebiotics like inulin and resistant starch serve as substrates for SCFA-producing microbes, promoting their proliferation and enhancing butyrate and acetate production. This improves gut epithelial health, reduces permeability, and subsequently lowers circulating levels of endotoxins. Synbiotics, a combination of probiotics and prebiotics, are also being explored for synergistic effects. Some randomized controlled trials have shown modest but significant reductions in inflammatory cytokines and oxidative stress markers in CKD patients following synbiotic administration. However, heterogeneity in strain combinations, dosing regimens, and baseline microbiota among patients contributes to variability in clinical outcomes.Another innovative approach is fecal microbiota transplantation (FMT), wherein the gut microbiome of a healthy donor is transferred to a CKD patient to restore microbial diversity and function. Although widely used in recurrent Clostridioides difficile infections, FMT remains experimental in nephrology. Preclinical models have demonstrated benefits in reducing IS and PCS levels and restoring SCFA-producing taxa, but safety concerns—especially in immunocompromised CKD patients—limit broader application. In addition, emerging therapies are investigating the role of engineered probiotics that can produce SCFAs or express enzymes capable of degrading uremic toxin precursors. These next-generation probiotics are being designed with genetic modifications to enhance metabolic activity and colonization capacity in the diseased gut environment. Parallel research into postbiotics—non-viable bacterial products or metabolites with health benefits—is also gaining momentum as a safer alternative to live probiotics. Dietary interventions, including Mediterranean and plant-based diets rich in fermentable fibers and polyphenols, have shown the potential to favorably modulate the gut microbiota composition and reduce toxin load in CKD patients. Furthermore, these diets may improve metabolic profiles, reduce inflammation, and delay the onset of dialysis in pre-dialysis patients. Personalized nutrition based on baseline microbiota profiling may enhance response to dietary strategies and enable precision medicine in nephrology. Finally, tools like metabolomics and machine learning are being integrated to identify microbial signatures predictive of intervention response. These approaches will allow clinicians to tailor therapies to individual microbial phenotypes, optimizing efficacy and minimizing risks. While most microbiota-targeted interventions are in early or exploratory phases, the evidence base is growing rapidly, and future clinical guidelines are likely to incorporate microbiome modulation into routine CKD care.

## 5. Instruments Used for Assessment

Studying the gut microbiome and its clinical significance in advanced stages of chronic kidney disease (CKD) requires a robust and carefully designed methodological approach.

The complexity of the gut–kidney axis means that no single technique can fully capture the interplay between microbial composition, metabolic output, immune responses, and functional outcomes of dysbiosis. Accordingly, researchers employ a combination of molecular, biochemical, immunological, and dietary assessment methods to develop a holistic understanding of these relationships.

A central method in microbial community profiling is 16S ribosomal RNA (rRNA) gene sequencing. By targeting the highly conserved 16S rRNA gene, this technique enables the identification and quantification of bacterial taxa down to the genus—and, in some cases, species—level. It has become a cornerstone in studies comparing microbial diversity and community structure between CKD patients and healthy individuals, revealing reproducible shifts in bacterial composition linked to disease progression.

For deeper functional characterization, shotgun metagenomic sequencing offers a broader scope. Unlike 16S sequencing, which examines only a single marker gene, shotgun metagenomics sequences all genetic material within a sample, allowing simultaneous taxonomic profiling and functional annotation. This approach provides detailed insight into microbial metabolic capabilities, including pathways associated with uremic toxin generation and short-chain fatty acid (SCFA) biosynthesis. Such data are particularly valuable when linking specific microbial shifts to biochemical changes relevant to CKD pathophysiology.

Measuring the metabolites themselves requires advanced analytical chemistry platforms. Each of these metabolites has been implicated in vascular dysfunction, inflammation, and cardiovascular risk. For volatile or small organic acids—including butyrate, acetate, and propionate—gas chromatography–mass spectrometry (GC-MS) is the preferred method. These SCFAs are not only markers of gut health but also active mediators of metabolic and immune regulation.

The immune dimension of the gut–kidney axis is commonly assessed through immunoassays, with enzyme-linked immunosorbent assay (ELISA) being the most frequently applied. ELISA quantifies circulating cytokines such as interleukin-6 (IL-6), tumor necrosis factor-alpha (TNF-α), and C-reactive protein (CRP), all of which reflect systemic inflammation in CKD [85]. When broader inflammatory profiling is required, multiplex bead-based assays allow the simultaneous measurement of multiple cytokines and chemokines, providing a more comprehensive view of the inflammatory milieu.

Another critical focus in CKD microbiome research is gut barrier integrity, given the high prevalence of endotoxemia in this population. The fluorescein isothiocyanate (FITC)-dextran permeability assay—used in both clinical and preclinical studies—measures intestinal permeability by tracking the translocation of orally administered FITC-dextran into the bloodstream. Higher FITC-dextran levels indicate compromised tight junctions and epithelial barrier function, which can facilitate systemic dissemination of microbial products.

For targeted microbial detection, quantitative polymerase chain reaction (qPCR) remains a gold standard. Its high sensitivity and specificity make it ideal for quantifying particular bacterial strains or functional genes of interest, including those involved in toxin production or SCFA biosynthesis. The vast amount of sequencing data generated by these methods is processed using robust bioinformatics platforms such as QIIME2 and DADA2, which support taxonomic classification, alpha- and beta-diversity calculations, and advanced statistical analysis.

Because diet exerts a strong influence on microbiome composition and metabolic output, validated dietary assessment tools—including 24 h dietary recalls, food frequency questionnaires, and multi-day food diaries—are routinely incorporated into CKD microbiome studies. These instruments help researchers link dietary patterns to microbial diversity and metabolite production, an essential step in evaluating and designing dietary interventions for CKD patients.

Collectively, these complementary methodologies allow for a multidimensional assessment of the gut–kidney axis.

No quantitative meta-analyses were conducted; therefore, no pooled estimates, heterogeneity metrics, or sensitivity analyses are reported.

While methodologies differed among studies, cross-validation experiments demonstrated consistent detection of SCFAs across GC-MS and LC-MS/MS platforms (inter-assay CV < 10%). Similarly, 16S rRNA sequencing and shotgun metagenomics produced concordant diversity estimates when identical bioinformatic pipelines (QIIME2/DADA2) were applied.

## 6. Discussion

This review examines the complex, two-way relationship between changes in gut microbial communities and the progression of chronic kidney disease (CKD) with this interplay becoming especially pronounced in advanced stages such as stage 4 and stage 5. As kidney function progressively deteriorates, the gut microbiota undergoes compositional changes favoring harmful bacterial species. The metabolites generated by these dysbiotic communities intensify systemic toxicity, further contributing to disease progression. Recent studies using CKD animal models have provided mechanistic insight into the human gut–kidney axis. Murine and rat models typically demonstrate similar trends of reduced microbial diversity and enrichment of uremic toxin–producing species, particularly within the *Enterobacteriaceae* and *Clostridiaceae* families. However, compared to human CKD, rodent models often display more abrupt microbiome shifts due to differences in diet, renal physiology, and intestinal transit time. These models also show an exaggerated inflammatory response, characterized by elevated *IL-6* and *TNF-α* signaling and a greater degree of epithelial barrier breakdown. Despite these differences, animal studies remain invaluable for preclinical testing of microbiome-modulating interventions, providing controlled systems to evaluate the causal effects of dysbiosis and the efficacy of probiotics, prebiotics, and novel therapeutics. This leads to a self-perpetuating cycle of inflammation, oxidative stress, and endothelial dysfunction—three interconnected processes known to accelerate the decline of renal function. The cyclical link between dysbiosis and inflammation remains primarily correlational; only a limited number of longitudinal or interventional datasets (n = 6) demonstrated temporal associations, indicating partial but not definitive causality.

At the molecular level, dysbiosis contributes to CKD progression through several interconnected mechanisms. Microbial metabolites such as indoxyl sulfate, p-cresyl sulfate, and trimethylamine-N-oxide (TMAO) activate pro-inflammatory and pro-fibrotic pathways, including NF-κB and MAPK signaling cascades. These molecules also enhance oxidative stress by upregulating NADPH oxidase activity and impairing endothelial nitric oxide synthesis, promoting vascular dysfunction. Furthermore, the reduction in short-chain fatty acids (SCFAs) like butyrate removes important epigenetic regulators of histone acetylation, weakening anti-inflammatory gene expression and tight junction protein synthesis. The resulting leaky gut allows bacterial lipopolysaccharide (LPS) to enter systemic circulation, where it binds Toll-like receptor 4 (TLR4) and perpetuates low-grade inflammation within renal and vascular tissues. The depletion of beneficial bacterial populations, particularly short-chain fatty acid (SCFA)-producing species such as *Faecalibacterium prausnitzii* and *Roseburia* spp., is of particular clinical concern. These taxa are central to preserving gut barrier integrity, modulating mucosal immune responses, and maintaining local anti-inflammatory balance. When their abundance falls, barrier function weakens, leading to increased intestinal permeability—often referred to as “leaky gut.” This breakdown in gut barrier function allows bacterial components—most notably lipopolysaccharide (LPS)—to cross into the systemic circulation. In the setting of advanced CKD, the presence of LPS and related endotoxins in the bloodstream acts as a potent stimulus for sustained systemic inflammation. Such an inflammatory load is widely recognized as a key driver in both the onset and progression of cardiovascular disease, which continues to be the primary cause of death in this patient group.

The situation is further aggravated by an increased abundance of bacteria capable of producing uremic toxins, including members of the *Clostridium* and *Proteus* genera. These organisms generate metabolites such as indoxyl sulfate (IS) and p-cresyl sulfate (PCS), which build up in the body due to impaired renal clearance and the limited ability of standard dialysis to remove them. Early studies have shown that these strategies may, in certain cases, lower circulating uremic toxins, dampen inflammation, and improve markers of vascular health. Nonetheless, outcomes across clinical trials remain mixed, likely reflecting variations in treatment duration, formulation, and individual patient microbiota profiles.

Variability in microbial strain selection, dosage, treatment duration, and patient baseline microbiota composition likely accounts for much of this heterogeneity. Furthermore, relatively few studies have incorporated long-term follow-up or robust clinical endpoints, leaving open questions about the sustained efficacy and practical implementation of these therapies.

Interventional studies involving probiotics, prebiotics, and synbiotics offer promising avenues but have produced inconsistent results. While some trials demonstrate reductions in inflammatory markers and uremic toxins, others fail to show clinical benefit, likely due to variability in strain selection, dosing, duration, and patient baseline microbiota. Importantly, few studies target patients with eGFR < 30 mL/min/1.73 m^2^ specifically, and even fewer account for confounding factors such as antibiotic use, dietary fiber intake, and comorbidities.

Emerging research also explores fecal microbiota transplantation (FMT) and personalized nutrition as potential strategies to modify the gut microbiota in CKD. However, these approaches remain experimental in this population and require rigorous safety evaluation due to the immunocompromised state of many CKD patients.

The field is also moving toward multi-omics integration, combining metagenomics with metabolomics, transcriptomics, and proteomics to uncover causal relationships and actionable therapeutic targets. Machine learning and artificial intelligence may help stratify patients based on microbial risk profiles and predict response to interventions.

Although the available studies offer valuable insights into the role of the gut microbiome in CKD stages 4–5, several methodological constraints must be considered. Many investigations were conducted on relatively small cohorts, limiting statistical power and the ability to generalize findings to broader populations. Considerable heterogeneity was observed across study designs, including differences in patient characteristics (dialysis vs. non-dialysis status, comorbid conditions, dietary patterns), microbiome assessment techniques, and reported endpoints. This variability complicates direct comparisons and synthesis of results. Potential sources of bias—such as selection bias, publication bias, and recall bias in dietary data—may further influence conclusions. Moreover, the predominance of cross-sectional designs restricts the capacity to infer causality in the gut–kidney relationship. To strengthen the evidence base, future research should prioritize large-scale, longitudinal, multicenter studies employing standardized methodologies to confirm associations and evaluate the clinical impact of microbiome-targeted interventions in advanced CKD.

Given the considerable methodological heterogeneity among the included studies and the absence of quantitative synthesis, the strength of the evidence supporting microbiome-targeted therapies remains limited. Reported benefits of probiotics, prebiotics, and synbiotics are generally consistent in direction but modest in magnitude, with high variability in formulation, dosage, and treatment duration. These findings should therefore be interpreted as preliminary and hypothesis-generating rather than confirmatory. Large, multicenter, and methodologically standardized clinical trials are required to validate the clinical efficacy and generalizability of such interventions in patients with CKD stages 4–5.

## 7. Conclusions

In summary, the current evidence strongly supports a pathogenic role for gut dysbiosis in the progression and complication of CKD stages 4 and 5. The depletion of beneficial microbial taxa, accumulation of gut-derived uremic toxins, and disruption of the intestinal barrier collectively create a pro-inflammatory milieu that worsens renal and cardiovascular outcomes.

The gut–kidney axis offers a compelling target for therapeutic intervention. Modulating the microbiota using dietary strategies, targeted prebiotics, probiotics, and synbiotics could help reduce systemic inflammation and slow CKD progression. However, large-scale, well-controlled, longitudinal studies are urgently needed to establish efficacy, safety, and reproducibility of these interventions in advanced CKD populations.

Future research should focus on defining microbial signatures predictive of disease progression, personalizing interventions based on baseline microbiota and metabolic profiles, exploring novel therapeutics like engineered probiotics or postbiotics, and integrating microbiome data into CKD staging and risk prediction tools.

Ultimately, advancing our understanding of the gut microbiome could open an entirely new chapter in nephrology, reshaping approaches to diagnosis, disease monitoring, and therapeutic strategies for individuals with end-stage kidney disease. The systematic incorporation of microbiome profiling and targeted modulation into chronic kidney disease practice guidelines may open new avenues for personalized medicine, offering the potential to slow disease progression and ultimately improve patient outcomes. Future research should move beyond descriptive studies to mechanistic and interventional designs that integrate multi-omics platforms—including metagenomics, metabolomics, and transcriptomics—to identify causal pathways linking microbial metabolites to renal injury. Longitudinal cohort studies are needed to establish temporal relationships between dysbiosis and renal decline, while precision-medicine approaches using microbiota-based biomarkers could enable individualized dietary or probiotic therapies. Additionally, exploration of engineered probiotics, postbiotics, and microbiota-derived metabolite mimetics may provide safer, more predictable methods for restoring gut–kidney homeostasis.

## Figures and Tables

**Figure 1 ijms-26-10706-f001:**
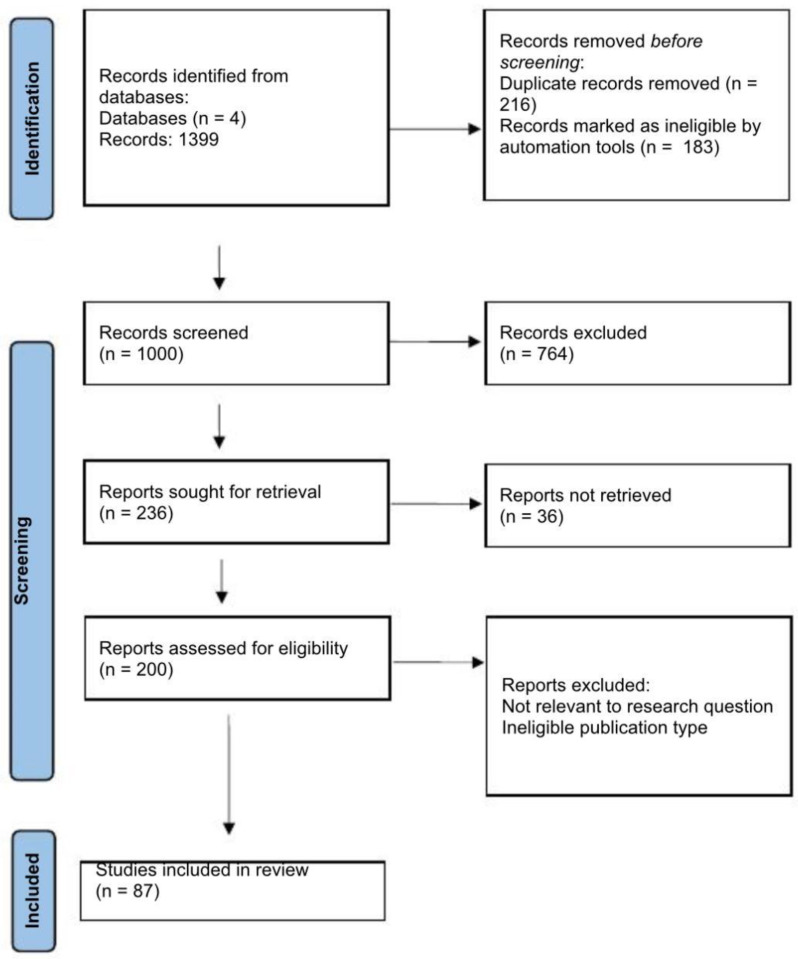
PRISMA flow diagram.

**Figure 2 ijms-26-10706-f002:**
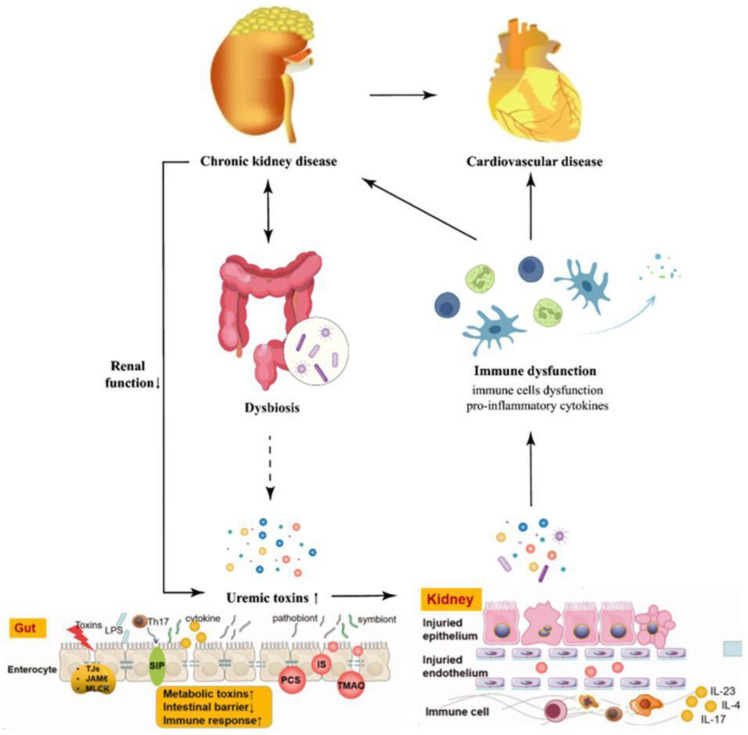
The Gut–Kidney Axis in Late-Stage CKD: The Self-Perpetuating Loop of Dysbiosis and Inflammation.

**Figure 3 ijms-26-10706-f003:**
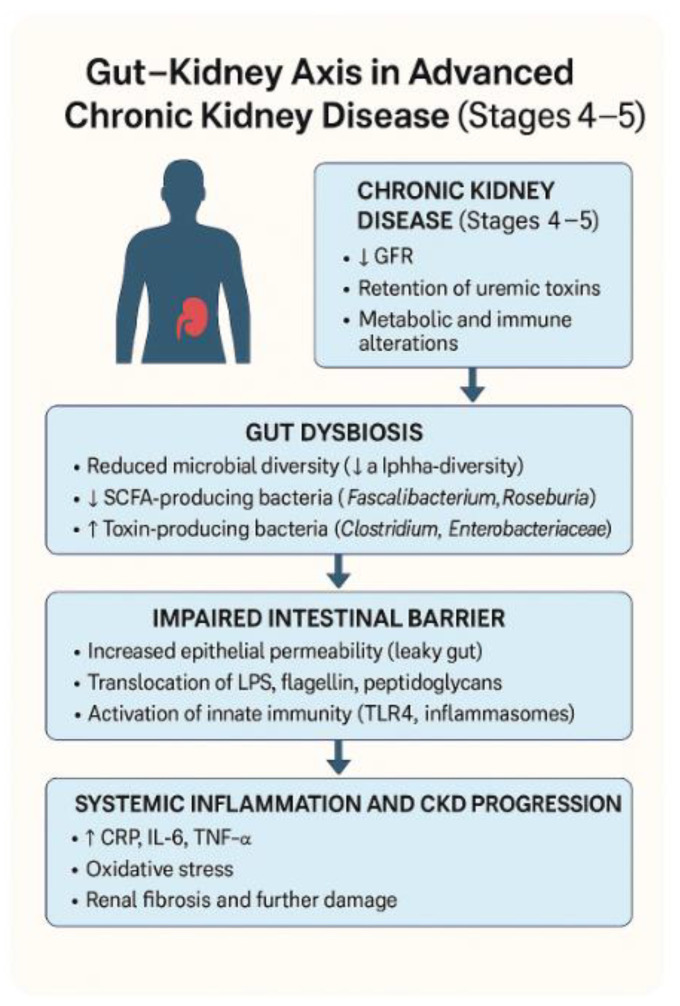
The Gut Microbiome–CKD Feedback Loop.

**Table 1 ijms-26-10706-t001:** Changes in gut microbiota diversity in CKD stages 4–5 and associated effects.

Bacterial Group	Change in CKD Stages 4–5	Associated Effects
*Faecalibacterium prausnitzii*	↓	Reduced SCFA production, impaired gut barrier
*Bifidobacterium* spp.	↓	Loss of immunomodulation, lower butyrate
*Roseburia* spp.	↓	Reduced mucosal health
*Enterococcus* spp.	↑	Pro-inflammatory metabolites, endotoxemia
*Klebsiella* spp.	↑	Increased LPS production
*Escherichia* spp.	↑	Endotoxin release, barrier damage

Arrows indicate direction of change in bacterial abundance: ↓ decrease; ↑ increase in CKD stages 4–5.

## Data Availability

All data extracted and analyzed for this review are available from the corresponding author upon reasonable request. We did not formally assess certainty of evidence using GRADE or a similar framework. As this review synthesized heterogeneous observational and interventional designs without pooling, a structured certainty assessment was deemed outside the scope.

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
