# Peer review of "Gut Microbiome in Patients with Chronic Kidney Disease Stages 4 and 5: A Systematic Literature Review"

_ijms, 2025, doi:10.3390/ijms262110706_

Round 1

Reviewer 1 Report

Comments and Suggestions for Authors

Minor comments:

This systematic review comprehensively evaluates 87 studies exploring gut microbiome alterations in patients with CKD stages 4 and 5. It highlights significant reductions in beneficial SCFA-producing bacteria and an overrepresentation of uremic toxin–generating microbes linked to systemic inflammation, barrier dysfunction, and cardiovascular risk. The review emphasizes the gut–kidney axis as a key pathogenic mechanism driving disease progression. Various interventions, including probiotics, prebiotics, synbiotics, and dietary modulation, are discussed as potential therapeutic strategies. However, the absence of meta-analysis and limited longitudinal data restrict the strength of the clinical conclusions.

(Lines 11–31): To what extent do the included studies represent global demographic and geographic variability in CKD populations, and how was selection bias minimized during data extraction?

(Lines 35–79): The introduction extensively discusses anticoagulant therapy and vitamin K antagonists. How is this content mechanistically or clinically integrated into the central theme of gut microbiome dysbiosis in CKD?

(Lines 80–127): The systematic review was not prospectively registered in PROSPERO or any equivalent database. What justifications are provided for this omission, and how might it impact methodological transparency and reproducibility?

(Lines 84–91): Were dialysis-dependent and non-dialysis patients analyzed as distinct subgroups to control for the potential confounding effects of dialysis on microbiome composition?

(Lines 125–127): The absence of a formal assessment for reporting bias is acknowledged. How does this limitation affect the interpretability and internal validity of the synthesized evidence?

(Lines 198–227): The manuscript reports reduced microbial alpha and beta diversity in CKD stages 4–5. What statistical indices (e.g., Shannon, Simpson) and significance thresholds were uniformly applied across studies to substantiate this conclusion?

(Lines 229–243): Given the emphasis on indoxyl sulfate (IS) and p-cresyl sulfate (PCS) accumulation, were quantitative comparative baselines provided for CKD versus healthy controls, and were these measurements standardized across included studies?

(Lines 244–260): Regarding intestinal barrier disruption, were alterations in tight junction proteins (occludin, claudin-1, ZO-1) assessed using validated quantitative methods such as immunoblotting, ELISA, or immunofluorescence imaging?

(Lines 261–277): The manuscript discusses a cyclical interaction between dysbiosis and inflammation. Were longitudinal or interventional datasets available to establish causality rather than correlation?

(Lines 282–311): Theoretical models such as the gut–kidney axis and endotoxemia–inflammation loop are described. Were these frameworks empirically derived from the reviewed studies, or conceptually synthesized from prior literature?

(Lines 314–352): Within the uremic toxin hypothesis, did the authors evaluate quantitative relationships between toxin levels (e.g., IS, PCS, TMAO) and renal function parameters such as eGFR or serum creatinine through meta-analytical correlation?

(Lines 354–396): Were analytical methodologies for quantifying short-chain fatty acids (SCFAs)—such as GC-MS or LC-MS/MS—consistent across the studies, ensuring comparability and reproducibility of the reported results?

(Lines 397–464): The review notes that dialysis may exacerbate endotoxemia. Did the included literature differentiate between hemodialysis and peritoneal dialysis modalities concerning their distinct microbiome alterations?

(Lines 465–522): Among the listed assessment techniques (16S rRNA sequencing, shotgun metagenomics, GC-MS, ELISA, FITC-dextran permeability assays), was any critical evaluation conducted comparing analytical sensitivity, specificity, or reproducibility?

(Lines 525–615): Considering the methodological heterogeneity and absence of quantitative synthesis, how robust and generalizable are the conclusions regarding the clinical efficacy of microbiome-targeted therapies such as probiotics, prebiotics, and synbiotics in CKD stages 4–5?

Author Response

Thank you for your response! Please see the file. Thank you!

Reviewer 2 Report

Comments and Suggestions for Authors

It is an interesting review article, however, contains less novelty things. The information presented by this manuscript were widely published every where, therefore, it sounds less attractive for readers.  

I have two suggestions, based on that, this review could bring important and novel information.

  1. Besides the stage 4 and 5, summarize the Stage 1 to stage 3, and make some preliminary analysis, tell readers the biggest dynamic changes between early stage and later stage. Several study comprehensive showed the early dynamic changes of gut microbiota, such as American Journal of Nephrology. 2023, 54(11-12):451-470.

  1. based on the CKD animal model, analyze and summarize the difference of gut dysbiosis between human CKD from later stage and murine or rat later CKD animal model. It is important for researcher to use animal model to try some intervention preclinical study. You can read the following publications: Communications Biology, 2023, 6, 1189. Journal of Pharmaceutical Analysis, 2024, 14(4):100931.

Author Response

Thank you for your response. Please see the file. Thank you!

Reviewer 3 Report

Comments and Suggestions for Authors

In this manuscript, the authors reviewed the role of gut microbiome in patients with chronic kidney disease.

The article is well written and interesting for readers, but I have a few minor suggestions:

  1. The authors should discuss in more detail the molecular mechanisms of the microbiota in the pathogenesis of chronic kidney disease. 
  2. Can modifying the microbiome provide significant therapeutic benefits?
  3. Future directions for microbiome research should be discussed in more detail.

Author Response

(The authors gave the same response as above.)
